# Development of Advanced Competencies in Physiotherapy: Impact of a Single-Blinded Controlled Trial on Ethics Competence

**DOI:** 10.3390/ijerph191710590

**Published:** 2022-08-25

**Authors:** Sara Cortés-Amador, Anna Arnal-Gómez, Elena Marques-Sule, David Hernández-Guillén, Catalina Tolsada-Velasco, Gemma V. Espí-López

**Affiliations:** 1Department of Physiotherapy, Faculty of Physiotherapy, University of Valencia, 46010 Valencia, Spain; 2UBIC Research Group, Department of Physiotherapy, University of Valencia, 46010 Valencia, Spain; 3Physiotherapy in Motion, Multispeciality Research Group (PTin MOTION), Department of Physiotherapy, University of Valencia, 46010 Valencia, Spain; 4Group of Physiotherapy in the Aging Process: Social and Health Care Strategies (PT_AGE), Department of Physiotherapy, Faculty of Physiotherapy, University of Valencia, 46010 Valencia, Spain; 5Exercise Intervention for Health (EXINH), Department of Physiotherapy, Faculty of Physiotherapy, University of Valencia, 46010 Valencia, Spain

**Keywords:** future physiotherapy professionals, advanced competencies, ethics, education, innovation, future healthcare professionals

## Abstract

Background: Innovation in the training of future physiotherapy professionals through the use of collaborative learning could be an effective method for developing advanced competencies such as professional ethics. This study aimed at comparing the effects of cooperative learning and individual learning on the knowledge of professional ethics, the perception of knowledge regarding professional ethics, the teaching quality assessment and satisfaction in future physiotherapy professionals. Methods: A prospective, assessor-blinded, controlled trial was performed. A 12-week program was carried out with future physiotherapy professionals. The cooperative learning group was based on group activities, while the individual learning group performed the same activities with an individual approach. Knowledge, perception of knowledge regarding professional ethics, teaching quality and satisfaction were assessed. Results: A total of 216 participants completed the study (cooperative group *n* = 106; individual group *n* = 110). The cooperative learning group showed higher knowledge and perception of knowledge regarding professional ethics compared to the individual learning group (*p* < 0.001 and *p* < 0.001, respectively). Additionally, the cooperative learning group reported higher scores in the teaching materials, attitude towards future professionals and the teacher’s global score. Conclusions: Cooperative learning showed a positive impact on developing advanced competencies such as knowledge and perception of knowledge regarding professional ethics. Both methodologies showed adequate results in the assessment of teaching quality and satisfaction.

## 1. Introduction

Innovation in the training of future professionals, specifically related to the development of advanced competencies such as professional ethics, is an important issue in healthcare. Physiotherapists, like other professionals in the health profession, are faced in their usual practice with ethical problems, the solving of which requires having an open, critical, tolerant and respectful attitude towards those who express different opinions or beliefs [1]. For this reason, physiotherapists must be prepared through their training to address situations such as the possible incompatibility between available resources and the patient’s needs, the management of conflicts between physiotherapists and other professionals, and respectfulness for the patient’s autonomy at all times [2,3,4]. As has been proven, ethical behavior and inter-professional working are key aspects of, and advanced competencies in, the healthcare environment [5]. Despite the available literature, it is noted that they are underdeveloped skills among recently graduated health professionals [6,7,8]; therefore, the application of innovative methodologies in the training of future professionals may be useful in healthcare.

Health ethics is defined as the set of principles, beliefs and values that guide health professionals when making medical care decisions [9,10,11]. Specifically, ethical competence is the sum of skills and knowledge acquired by the future professional that must be coordinated with professional values [12].

One of the objectives of the European Higher Education Area (EHEA) paradigm is for future physiotherapy professionals to be the centerpiece of the teaching–learning process. In this regard, it is necessary to promote the development of advanced competencies, such as professional ethics, and generate situations that allow the acquisition of knowledge, skills and interpersonal abilities for future healthcare professionals. Traditionally, the individualistic form of learning has been carried out, where there is no type of correlation between the achievement of the goals proposed for the future professionals. Each participant seeks his or her goal and their performance or attitude does not influence the others. Accordingly, each individual seeks his or her own benefit, without considering the other participant. However, in cooperative learning processes, the objectives of the participants are linked to each other, with a positive correlation between the achievement of their goals. The results of each member affect the rest of their peers [13].

Promoting innovative methodologies in the training of future healthcare professionals, such as cooperative learning (CL), can be a strategy that allows them to achieve ethical competence by creating situations that resemble a clinical context. This type of methodology promotes analytical and critical thinking, teamwork, generates positive emotional climates and favors respect for diversity [14]. Cooperative learning is defined as an approach in which participants are assigned to small structured groups and help each other to achieve common objectives, so that participants are not only responsible for learning the content themselves, but also for helping their peers in their learning process [15]. It is a method that differs from competitive or individualistic learning, in which each participant works on his or her own to achieve their own goals, thus separating themselves from their peers.

Learning is not only a cognitive process; it is also a social process. For this reason, CL is not only important to transform the way of learning itself, but also to promote social strategies among the participants [16]. A number of studies confirm that CL allows the development of interpersonal skills, better relationships between participants, the skills to listen, share ideas and collectively develop a new meaning and understanding, thereby generating an atmosphere of trust and respect [17,18,19]. The work by Laal and Ghodsi (2012) [20], when comparing cooperative learning and training based on individual effort and competitiveness, confirmed that CL translates into higher performance and higher productivity, helping to maintain more committed and supportive relationships and translating into better psychological health and greater self-esteem of individuals.

Therefore, teachers need to provide opportunities for future healthcare professionals to interact with each other, transforming master classes, in which there is no possibility for the future professional to interact, into spaces in which they not only have the opportunity to participate, but are also able to share and work on the content explained with the rest of their peers [21]. Therefore, implementing innovative training methodologies such as collaborative learning, and comparing it to individual learning, could yield positive results and be an effective method for developing advanced competencies such as professional ethics in future healthcare professionals.

Therefore, the aim of this study was to compare the effects of cooperative learning and individual learning on the knowledge of professional ethics, perception of knowledge regarding professional ethics, and assess teaching quality and satisfaction in future physiotherapy professionals.

## 2. Materials and Methods

### 2.1. Design and Setting

The study was a prospective, assessor-blinded, non-randomized controlled trial. Participants were assigned to a CL group (CLG, *n* = 106) or to an IL group (ILG, *n* = 110). The assessor was blinded to group allocation. Participants were reminded not to reveal their program group at the post-intervention examination. The blinded assessor collected all the baseline and post-intervention measures and entered the data. Participation in the study was voluntary and all participants were assessed under the same conditions before and after intervention. All the enrolled participants were informed of the purpose of the study and provided written informed consent.

### 2.2. Participants

A total of 241 participants were recruited from January 2019 to April 2019. They were voluntary future physiotherapy professionals from the physiotherapy degree program, aged between 18 and 30 years. All the enrolled participants were informed of the purpose and procedures of the study and provided written informed consent. The exclusion criterion was having had prior training on ethics. The study was carried out at the authors’ institution.

### 2.3. Intervention

All the interventions were performed at a university research laboratory. The interventions were delivered by a teacher, experienced in the trial procedures and with more than 10 years of experience in professional ethics. All the participants received a 12-week program, one session/week, with a total of ten theoretical sessions (1 h each) and two practical sessions (2 h each). The topics in both groups consisted of: moral values, ethics and morals, bioethics and professional ethics, the differences between ethical actions and legal actions, the dimensions of professional ethics, the professional values of physiotherapists, legal regulations, basic ethical principles, the competencies for a good deontological professional code, different ethical situations, the components of moral behavior, professional responsibility and other ethical questions and ethical theories. Both groups performed 10 h of theoretical sessions and 4 h of practical sessions.

#### 2.3.1. Cooperative Learning Group

Theoretical sessions: consisted of 5 participatory master classes, 1 h each, which generate more participation, interactivity, meaning of the information for the participant, and focus on them as the protagonist of their own knowledge, part of the concept of the learning process [22], involving the participant, for example through questions [23]. In addition, 1-h sessions with group activities were carried out, to favour the understanding of the analysed concepts, through the use of cooperative methodologies such as Four Corners, Aquarium, brainstorming, glossary, concept maps or Philips 6-6 [5].

Practical sessions: Two practical sessions were carried out, 2 h each. These sessions consisted of analysis of situations in which the ethics of the profession was compromised. Participants had to solve this situation in groups of four people using the method of analysis of ethical cases Realm-Individual Process-Situation (RIPS) [24]. The objective was to stimulate critical thinking and resolution of two ethical cases. Once the ethical case was resolved, the following strategies were carried out:

Reduced groups and guided debate, in which the resolution of ethical cases was shared. The objective of this technique was to carry out a deep and critical exploration of the cases, staying focused on building knowledge through collaboration [25].

Spontaneous group discussion, which was also used to share ethical cases [26]. The objective of this technique was to promote the resolution of cases among participants through spontaneous debate among the different groups [25].

Table A1 and Table A2 show the techniques used for each theoretical and practical session of the CLG and a detailed explanation of each technique.

#### 2.3.2. Individual Learning Group

Theoretical sessions: They consisted of 10 participatory master classes of 1 h each, addressing the same theoretical topics as in the CLG.

Practical sessions: Two sessions were held, lasting 2 h each, in which the individualized analysis of the same ethical situations as for the CLG was carried out, but each participant had to resolve it individually through the same method of the analysis of ethical cases (RIPS) [24]. Each participant was given two ethical cases to solve individually.

### 2.4. Outcomes

After giving informed consent, all the participants provided demographic information. They then underwent assessment, conducted by a trained teacher in managing the evaluation tools with more than 10 years’ experience in teaching professional ethics to future healthcare professionals.

#### 2.4.1. Knowledge of Professional Ethics

The participants’ knowledge was measured with an ad hoc multiple-choice questionnaire. The participants were not previously informed about the retention exams, to avoid any preparation for the test. The tests included 15 multiple-choice questions that assessed basic professional ethics-related concepts that future physiotherapy professionals are required to know, such as: moral values, bioethics and professional ethics in physiotherapy, professional ethical principles, the physiotherapists’ deontological code, ethical situations and professional values in physiotherapy.

#### 2.4.2. Perception of Knowledge Regarding Professional Ethics

A self-reported questionnaire including 19 items, (the Perceptions about Knowledge regarding Professional Ethics in Physiotherapy, PKPEPT, Cronbach’s alpha = 0.76), was previously described [21,25]. Higher scores indicated a better perception of knowledge. The questionnaire items included basic professional ethics-related concepts that future physiotherapy professionals are required to know. Therefore, the questions assessed participants’ perceived knowledge of the basic concepts in professional ethics and their application in physiotherapy care. The items were created based on their importance and being clearly related with the perception of knowledge.

The questionnaire included concepts related to professional ethics such as: moral values and non-moral values; ethics and morals; bioethics and professional ethics in physiotherapy; the ethical professional act and the legal professional act; professional ethical principles; the physiotherapists’ deontological code; ethical problems, ethical dilemmas, moral stress, moral temptation and silence in physiotherapy; competencies of a good professional; the professional values of physiotherapy—autonomy, beneficence, non-maleficence and justice; ethical problems that physiotherapists are faced with in their daily clinical practice; ethical principles of professional ethics applied to physiotherapy and established by the World Confederation of Physiotherapy; ethical principles and behavior codes; methods to analyze ethical problems and ethical dilemmas of the profession.

Items were written in a structured and comprehensive way with affirmative sentences, whereas adverbs such as “always” and/or “never” were avoided in order to facilitate comprehension and response. Items were in a four-point Likert format, with scores ranging from 1 (none) to 4 (high) [9,11,19,21]. Scores were obtained as follows: for each item, 1 point was assigned when the option “none” was selected, 2 points when the option “low” was selected, 3 points when the option “moderate” was selected and 4 points when the option “high” was selected. Thus, the sum of points for each participant was calculated, 76 points being the maximum possible score for the questionnaire.

#### 2.4.3. Teaching Quality Assessment

An online survey including different items that evaluated the teaching methodologies, materials and general satisfaction with the teaching quality was performed. Each item was scored on a Likert scale (from 1 “totally disagree” to 5 “totally agree”). This assessment was only conducted at the end of the program.

#### 2.4.4. Satisfaction Questionnaire

The CL group carried out an ad hoc satisfaction questionnaire, using the Kahoot application. The satisfaction questionnaire included 16 questions in total: 5 dichotomous questions about opinions on methodology, 5 dichotomous questions about the topics, 3 Likert-type questions (not at all/not much/quite a lot/a lot) about the interest in and usefulness of the content(s), and 3 Likert-type questions about satisfaction, usefulness and fun with respect to the techniques used, on a scale of 0 to 5 points. This evaluation was only carried out at the end of the program.

### 2.5. Statistical Analysis

The descriptive results of the continuous data were expressed as the mean and standard deviation, while the nominal data were described in frequencies and percentages. The Kolmogorov–Smirnov test revealed a normal distribution of the variables (*p* > 0.05) and the assumption of homoscedasticity was requested using Levene’s test. For the inferential analysis, the mean comparison (Student’s *t*-test and Pearson’s chi-square test) between the educational years was used for the independent samples. Two-tailed tests were always conducted and the significance level was set at *p* < 0.05. The statistical analysis was performed using SPSS v. 24.0 (SPSS Inc., Chicago, IL, USA, licensed from the authors’ institution). An external assistant not involved in the study performed the statistical analysis.

### 2.6. Ethical Considerations

The study protocol was approved by the Institutional Review Board of the University of Valencia, Spain (H1515588244257) and all procedures were conducted according to the principles of the Declaration of Helsinki (October 2013, Fortaleza, Brazil). This was an Educational Innovation Project of the authors’ University (UV-SFPIE_PID19-1095272). This trial was registered at www.clinicaltrials.gov (accessed on 1 May 2021) (registration number: NCT03795077).

## 3. Results

From the 241 initial candidates, a total of 216 future physiotherapy professionals agreed to participate and were analyzed (response rate 89.62%) (Figure 1). The average age of the participants was 21.06 ± 4.76 years; 109 were women (50.5%). A total of 99.6% of the sample wanted to work as a physiotherapist after finishing their studies and 35.6% had family members working in health professions. All of them agreed voluntarily to participate in the study. There were no statistical differences between the groups at the baseline. The demographic and clinical characteristics of the participants by group are depicted in Table 1.

### 3.1. Knowledge of Professional Ethics

Table 2 shows the results of the multiple-choice questionnaire. There were no significant differences at the baseline. After the program, the CLG increased their scores in the multiple-choice questionnaire with a significant difference in relation to the ILG (*p* < 0.001). In the within-group analysis, both groups significantly improved their knowledge of professional ethics at the end of the program compared to the baseline (CL: *p* < 0.001; IL: *p* < 0.001).

### 3.2. Perception of Knowledge Regarding Professional Ethics

Table 3 shows the total score in the PKPEPT questionnaire by group, and significant differences between the groups. The CLG had a significantly higher perception of their knowledge regarding professional ethics compared to the ILG (*p* < 0.001) after the teaching program. Moreover, both groups increased their perception of knowledge after 12 weeks, and the increase was statistically significant in the within-group analysis (CLG: *p* < 0.001; ILG: *p* < 0.001).

Regarding the response options of the PKPEPT, before the training, both groups showed that more than 25% and 40% of the sample answered that their perception of knowledge regarding professional ethics, was “none”, or “low”, respectively. After the teaching program, at least 50% of the sample in both groups answered “moderate”, highlighting that more than 30% of the CLG stated a “high” perception of ethical knowledge (Figure 2).

### 3.3. Teaching Quality Assessment

The teaching quality assessment showed that both groups generally rated the innovative training with high scores (the mean score of the survey for CLG: 4.53; ILG: 4.57). The CLG rated higher certain items related to the teaching materials (CLG: 4.63; ILG: 4.56), the attitude towards the participants (CLG: 4.72; ILG: 4.63) and the global score of the teacher (CLG: 4.47; ILG: 3.95). The ILG rated the teaching methodology with higher scores (CLG: 4.43; ILG: 4.58). The item related to general satisfaction was higher for the CLG (CLG: 4.81; ILG: 4.63).

### 3.4. Satisfaction Questionnaire

The CLG (100%) considered the methodology adequate for the learning process, and a high percentage of them thought it allowed them to achieve the objectives of the training activity (92.5%) and that it was better than traditional methodologies (86.8%). In addition, 87.7% considered that the activities helped to understand the concepts better and 67.0% had an overall satisfaction of 4 points out of 5 (Table 4).

## 4. Discussion

To the best of our knowledge, this is the first study that evaluates the effectiveness of CL on the development of advanced competencies such as the knowledge of professional ethics and the perception of knowledge regarding professional ethics, as well as assesses the teaching quality and satisfaction in future physiotherapy professionals.

At the end of the training, the CLG significantly improved their knowledge and perception of knowledge regarding professional ethics, when compared to the ILG.

In relation to their knowledge of professional ethics, upon completion of the teaching process, the CLG improved 2.42 points (from 5.34 to 7.76), while the ILG improved only 1 point (5.42–6.42) Knowledge acquisition is usually the ultimate end of the college experience. For these reasons, it is important that training activities attempt to maximize knowledge acquisition [20,27]. For García & Traver (2001) [13], CL is better than IL for the socializing function, learning and performance.

The results of the present work, as well as those obtained in the meta-analysis by Johnson (2013) [28], the findings reported by Mehra and Thakur (2008) [29], and Nichols (1996) [30] confirm that CL obtain higher scores than IL in knowledge. Hancock (2004) [31] confirms that an approach in which the participant shares ideas with the group increases the degree of participation. Active participation in the learning process is an important predictor of achievement. Our results confirm this hypothesis. Our innovative training proposal provided the opportunity for future physiotherapy professionals to share and exchange their opinions and, in turn, achieve better results. This has an effect on learning because, to the extent that participants have opportunities to explain the contents learned to other peers, the information will be more easily retained in memory and learning will be more effective [32]. Understanding how future physiotherapy professionals learn better to develop advanced competencies, such as ethics, is a key aspect to designing training proposals that promote these aspects in healthcare environments.

In relation to the perception of knowledge regarding professional ethics, the CLG showed a significantly higher score compared to the ILG in their perception of knowledge regarding professional ethics.

After the learning process, it was noted that only 0.5% of the CLG answered that their perception of knowledge was “none” while, in the ILG, the percentage was 1.44%. For the CLG, 86.24% of the responses were moderate (53.27%) and high (32.97) while, in the ILG, only 77.65 gave moderate (52.15) or high responses (25.50%). The results of the work carried out by Delany and Kulju support that the perception of knowledge of ethics is the prerequisite for ethical competence. That is, to the extent that the future healthcare professional has greater knowledge about the content of ethics, there is a greater probability that he or she will be able to resolve/cope with professional ethical issues [3,33].

In this regard, we consider that the validation of a questionnaire applicable to training models of ethical competencies in physiotherapy, which can assess the perception of participants about their knowledge on the subject, is essential for good professional practice. However, we have verified that there are very few publications on validation of the questionnaires related to professional ethics in physiotherapy and, although a growing interest in the subject is beginning to emerge [34], we have found no studies that have used validated instruments to measure the perception of knowledge in future physiotherapy professionals. Given such evidence, and in accordance with the appreciation of various authors who consider that teaching ethics to future physiotherapists is essential [35], we highlight the importance of using a validated questionnaire that assesses perceived knowledge of professional ethics in physiotherapy.

On the other hand, when the results on the teaching quality assessment were analyzed, the ILG scored 0.15 points higher than the CLG in teaching methodology. However, the item relating to general satisfaction was 0.18 higher for the CLG than for the ILG. This may be due to a lack of cooperative skills, communication skills or interpersonal skills, which may decrease participation in cooperative activities [36] and therefore positively score for ILG. It is important that teachers be aware of the difficulties and challenges that CLGs pose for future professionals, so that training allows them to develop the necessary skills for cooperation.

Regarding the satisfaction questionnaire, all the participants adequately valued the learning methodology and 92% believed that the methodology was better than traditional methodologies based mainly on master classes. Overall, our results show that both CLG and ILG have a positive impact on teaching quality and satisfaction. Therefore, it would be interesting if the design of the degree programs included activities with both an individualized and a cooperative approach, as both methodologies can complement each other and be equally effective, as shown by our results.

### Limitations and Strengths

Among the limitations of the present work, it should be noted that no analysis was conducted on the possible changes in the relationships between participants once the teaching process has finished. In the same way, conducting a follow-up with participants to analyze whether learning was sustained over time could yield interesting results. In addition, all the students were from the same institution; thus, it could be a bias since it is difficult to avoid students talking to each other and sharing information about the contents. As such, our results may not be applicable globally and should be interpreted cautiously. The analysis on the impact of the different educational methodologies in the field of physiotherapy is still scarce; however, this study offers a working tool so that teachers can promote the development of professional ethics, a fundamental aspect for the profession of physiotherapy and in the health area, in general.

## 5. Conclusions

Future physiotherapy professionals can develop advanced competencies such as professional ethics by means of cooperative learning. The knowledge of professional ethics and perception of knowledge regarding professional ethics can be improved using cooperative learning in future professionals. The future professionals in this study considered that the teaching quality and satisfaction were adequate when using both individual and cooperative learning.

## Figures and Tables

**Figure 1 ijerph-19-10590-f001:**
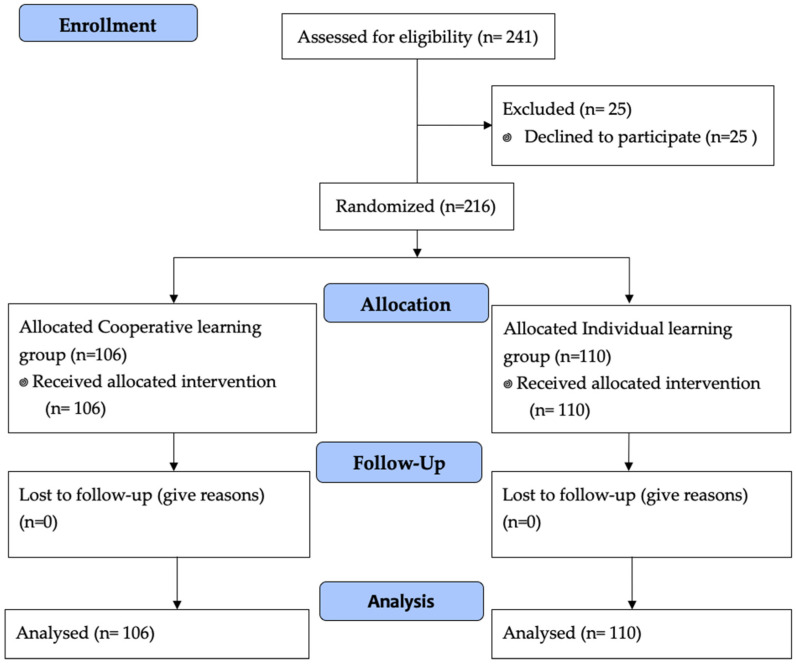
Flow diagram of study participation.

**Figure 2 ijerph-19-10590-f002:**
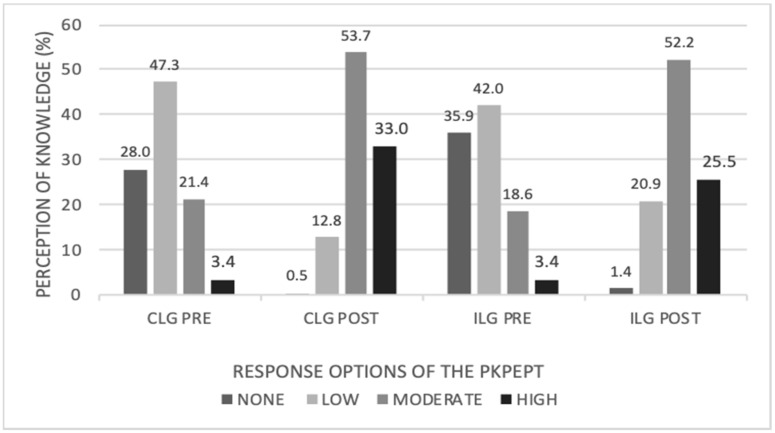
Number of participants that answered each score of the PKPEPT questionnaire, by group, before and after the innovative training methodologies. PKPEPT: Perception of Knowledge; CLG: cooperative learning group; ILG: individual learning group.

**Table 1 ijerph-19-10590-t001:** Sociodemographic characteristics of the sample.

	Total *n* = 216	CL *n* = 106	IL *n* = 110	*p*-Value
Age (mean ± SD)	21.06 ± 4.76	20.84 ± 4.04	21.26 ± 5.38	0.51
Gender, *n* (%):				0.13
*Male*	107 (49.5)	47 (44.3)	60 (54.5)	
*Female*	109 (50.5)	59 (55.7)	50 (45.5)	
Do you have family members who work in the health area?, *n* (%)	77 (35.6)	38 (35.8)	39 (35.5)	0.95
Did you study physiotherapy voluntarily?, *n* (%)	216 (100)	106 (100)	110 (100)	N.A.
Do you want to work as in physiotherapy in the future?, *n* (%)	215 (99.6)	106 (100)	109 (99.1)	0.33

CL: cooperative learning; IL: individual learning; SD: Standard deviation.

**Table 2 ijerph-19-10590-t002:** Knowledge of professional ethics before and after the teaching programs for both groups.

Multiple-Choice Questionnaire *	Pre-Intervention (Mean ± SD)	Post-Intervention (Mean ± SD)	*p*-Value within Groups
**CLG** ** *n* ** ** = 106**	5.34 ± 1.17	7.76 ± 1.26	<0.001
**ILG** ** *n* ** ** = 110**	5.42 ± 1.17	6.14 ± 1.25	<0.001
** *p* ** **-value between groups**	0.662	<0.001	

CLG: cooperative learning group; ILG: individual learning group; SD: Standard deviation. * Score over 10 points.

**Table 3 ijerph-19-10590-t003:** Perception of knowledge regarding professional ethics after the training interventions in the studied groups.

	Pre-PKPEPT * (Mean ± SD)	Post-PKPEPT * (Mean ± SD)	*p*-Value within Groups
**CLG** ** *n* ** ** = 106**	19.03 ± 7.77	41.68 ± 6.76	<0.001
**ILG** ** *n* ** ** = 110**	17.02 ± 7.26	38.27 ± 7.73	<0.001
** *p* ** **-value between groups**	0.06	<0.001	

CLG: cooperative learning group; ILG: individual learning group. PKPEPT: Perception of Knowledge regarding Professional Ethics in Physiotherapy Questionnaire; SD: Standard deviation. * Score over 76 points.

**Table 4 ijerph-19-10590-t004:** Answers of the satisfaction questionnaire at the end of the program for the cooperative learning group.

Satisfaction Questionnaire						CLG *n* = 106
**(1) Methodology**						
I positively value the learning methodology used, *n* (%)						106 (100)
I believe that the methodology used has allowed me to achieve the objectives of the training activity, *n* (%)						98 (92.5)
I believe that the methodology is better than those based mainly on master classes, *n* (%)						92 (86.8)
The use of this methodology has had positive results for me, *n* (%)						92 (86.8)
The teaching methodology has improved my ability to accept other classmates’ proposals even if they are different from mine, *n* (%)						78 (73.6)
**(2) Topics**						
The activities have helped to understand the concepts better, *n* (%)						93 (87.7)
The extension of each topic has been enough to understand the concepts, *n* (%)						91 (85.8)
In general, the contents seem specific enough for future physiotherapy professionals, *n* (%)						83 (78.3)
The language used in the topics is written in a comprehensive way, *n* (%)						80 (75.5)
I would recommend my peers to study these topics, *n* (%)						72 (67.9)
**(3) Contents**			**Low**	**Little**	**Enough**	**High**
How appropriate has the combination of virtual activities and theoretical and practical contents been?, *n* (%)			1 (0.9)	9 (8.5)	79 (74.5)	17 (16.0)
How useful have the topics discussed in the theory and practice classes been?, *n* (%)			1 (0.9)	7 (6.6)	88 (83.0)	10 (9.4)
How interesting have the topics discussed in the theory and practice classes been?, *n* (%)			4 (3.8)	14 (13.2)	83 (78.3)	5 (4.7)
**(4) Satisfaction, utility and fun**	**0**	**1**	**2**	**3**	**4**	**5**
Level of satisfaction, *n* (%)	0 (0)	0 (0)	1 (0.9)	18 (17.0)	71 (67.0)	16 (15.1)
Level of utility, *n* (%)	0 (0)	0 (0)	1 (0.9)	17 (16.0)	62 (58.5)	26 (24.5)
Level of fun, *n* (%)	0 (0)	6 (5.7)	15 (14.2)	50 (47.2)	29 (27.4)	6 (5.7)

CLG: cooperative learning group.

## Data Availability

Not applicable.

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
