# Peer review of "Development of Advanced Competencies in Physiotherapy: Impact of a Single-Blinded Controlled Trial on Ethics Competence"

_ijerph, 2022, doi:10.3390/ijerph191710590_

Round 1

Reviewer 1 Report

Congratulations on the article, it is very interesting.

I will now proceed to the doubts read in the article:

When they comment on the division of groups by flowchart they should follow the PRISMA method as this would explain it better, as it does not reflect which is which of the groups and why the division is 106vs110.

As for the statistical analysis:

When you talk about normality test, first you will have to perform the KOLMOGOROV-SMIRNOF test, since your sample is larger than 50, then according to the result of p value, we will know which test to perform, and then perform the Levene test (test that is used to know if there is or not homogeneity of variances).

Author Response

Reviewer#1:

General comments: Congratulations on the article, it is very interesting.

Response: We appreciate your comment.

Comment 1: When they comment on the division of groups by flowchart they should follow the PRISMA method as this would explain it better, as it does not reflect which is which of the groups and why the division is 106vs110.

Response 1: We thank the reviewers for this suggestion. We have changed the figure in order to adapt it to CONSORT guidelines for clinical trials, since PRISMA is usually used for reviews. In addition, we have improved the explanation of the groups.

Comment 2: As for the statistical analysis: When you talk about normality test, first you will have to perform the KOLMOGOROV-SMIRNOF test, since your sample is larger than 50, then according to the result of p value, we will know which test to perform, and then perform the Levene test (test that is used to know if there is or not homogeneity of variances).

Response 2: We totally agree with you and thank you for your advice. We have revised it in the manuscript.

Changes in the text: “The Kolmogorov-Smirnov test revealed a normal distribution of the variables (p>0.05) and the assumption of homoscedasticity was requested using Levene’s test.”

Reviewer 2 Report

Interesting study with future physiotherapists. However it seems that both methods are good even the colaborative is slightly better.

Minor corrections are needed and two questions.

Corrections:

On line 63 i believe this is the first time that the acronym EHEA apears. So, it should appear in full.

Please explain or correct the confusion about the hours of practical sessions since in line 129 appears as 3h but after that it seems that practical classes are of 2h... (line 134-135, 144, 161...)

I have two questions:

How have authors controlled the effect of all the students are from the same institution. This could be a bias since you can't stop students from talking to each other and our experience tell us that most of their conversations at school are about the trachers and the content taught. This could influence also the assessment of the quality of teaching by both groups.

Other question is in fact, is obvious that students must know more about a subject after classes on that subject. So. differences between before and after training should obviously exist. However authors do not discuss enough (in my opinion) why after teaching differences were not significant.

Author Response

Reviewer#2:

General comments: Interesting study with future physiotherapists. However, it seems that both methods are good even the collaborative is slightly better.

Response: Thanks to the reviewer for approving our idea. We will be very happy to edit the text further and answer your query, based on helpful comments from you.

Comment 1: On line 63 I believe this is the first time that the acronym EHEA appears. So, it should appear in full.

Response 1: You are right, sorry for the mistake. It has been written correctly.

Changes in the text: “… of the European Higher Education Area (EHEA) paradigm…”

Comment 2: Please explain or correct the confusion about the hours of practical sessions since in line 129 appears as 3h but after that it seems that practical classes are of 2h... (line 134-135, 144, 161...)

Response 2: We are sorry about the confusion. We made a typo in line 129-130.

Changes in the text: “…2 practical sessions (2 hours each)."

Comment 3: How have authors controlled the effect of all the students are from the same institution. This could be a bias since you can't stop students from talking to each other and our experience tell us that most of their conversations at school are about the trachers and the content taught. This could influence also the assessment of the quality of teaching by both groups.

Response 3: Thank you very much for your comment. We have proceeded to reflect this in the limitations of the study.

Changes in the text: “… interesting results. In addition, all the students were from the same institution, thus it could be a bias since it is difficult to avoid that students talk to each other and share information about the contents. As such our results may not be applicable globally and should be interpreted cautiously.”

Comment 4: Other question is in fact, is obvious that students must know more about a subject after classes on that subject. So. differences between before and after training should obviously exist. However, authors do not discuss enough (in my opinion) why after teaching differences were not significant.

Response 4: We are sorry about the mistake we performed in relation to the tables. The information included in the text of the manuscript was correct, although the information included in tables 2 and 3 was not correct. We have changed tables 2 and 3 in order to clarify the results. Then Tables 2 and 3 show, in short, the following information: both groups (cooperative learning and individual learning) improved their knowledge and perception of knowledge after the intervention. In addition, it is observed that there were between-group differences after the intervention.

Reviewer 3 Report

It is the first study that evaluates the effectiveness of cooperative learning (CL) on the development of advanced competencies such as knowledge of professionals ethics, perception of knowledge regarding professional ethics, as well as teaching quality assessment, and satisfaction, in future physiotherapy professionals.

The Authors showed a positive impact of CL in developing advanced  competencies such as knowledge and perception of knowledge regarding professional ethics

The Authors have presented sufficient data. The appropriate tables and figures have been provided. The article is easy to read and logically structured.  The methods are adequately described. The authors used appropriate statistical methods. The conclusions are consistent with the presented evidence and arguments.

the strength of this paper: very interesting topic; material and methods-the right choice of methodology methods, which were presented incomprehensible way; the obtained results are presented in the form of figures and tables, which are clear and easy to understand; the discussion- supports the results properly and refers to the current literature inappropriate manner; the conclusions- based on the obtained results, they are consistent with evidence and arguments. They address the main question posed.  The Authors used appropriate references.

Author Response

Reviewer#3

General comments: It is the first study that evaluates the effectiveness of cooperative learning (CL) on the development of advanced competencies such as knowledge of professionals’ ethics, perception of knowledge regarding professional ethics, as well as teaching quality assessment, and satisfaction, in future physiotherapy professionals.

The Authors showed a positive impact of CL in developing advanced competencies such as knowledge and perception of knowledge regarding professional ethics

The Authors have presented sufficient data. The appropriate tables and figures have been provided. The article is easy to read and logically structured.  The methods are adequately described. The authors used appropriate statistical methods. The conclusions are consistent with the presented evidence and arguments.

the strength of this paper: very interesting topic; material and methods-the right choice of methodology methods, which were presented incomprehensible way; the obtained results are presented in the form of figures and tables, which are clear and easy to understand; the discussion- supports the results properly and refers to the current literature inappropriate manner; the conclusions- based on the obtained results, they are consistent with evidence and arguments. They address the main question posed.  The Authors used appropriate references.

Response: We thank the reviewer for his/her review and comments.